# Long-term exposure to ambient PM2.5, particulate constituents and hospital admissions from non-respiratory infection

Yijing Feng [1] ✉, Edgar Castro [2], Yaguang Wei[2], Tingfan Jin [2], Xinye Qiu [2], Francesca Dominici [3] & Joel Schwartz [1,2]

The association between PM2.5 and non-respiratory infections is unclear. Using data from Medicare beneficiaries and high-resolution datasets of PM2.5 and its constituents across 39,296 ZIP codes in the U.S between 2000 and 2016, we investigated the associations between annual PM2.5, PM2.5 constituents, source-specific PM2.5, and hospital admissions from non-respiratory infections. Each standard deviation (3.7-µg m−3) increase in PM2.5 was associated with a 10.8% (95%CI 10.8–11.2%) increase in rate of hospital admissions from non-respiratory infections. Sulfates (30.8%), Nickel (22.5%) and Copper (15.3%) contributed the largest weights in the observed associations. Each standard deviation increase in PM2.5 components sourced from oil combustion, coal burning, traffic, dirt, and regionally transported nitrates was associated with 14.5% (95%CI 7.6–21.8%), 18.2% (95%CI 7.2–30.2%), 20.6% (95%CI 5.6–37.9%), 8.9% (95%CI 0.3–18.4%) and 7.8% (95%CI 0.6–15.5%) increases in hospital admissions from non-respiratory infections. Our results suggested that non-respiratory infections are an under-appreciated health effect of PM2.5.

It is estimated that 15% of all the deaths across the world are directly attributable to infectious disease each year[1]. While respiratory infections draw the most attention, non-respiratory infections are also an important health burden. For example, in 2015, it was estimated that the healthcare expenditure for intestinal infections was $6.4 billion in the United States[2]. There were around 10.5 million office visits for urinary tract infection, causing $3.5 billion in societal costs per year[3,4].

Fine particulate matter (PM2.5) is a well-recognized risk factor for health and has been found to be associated with multiple adverse health outcomes such as cardiovascular disease and respiratory disease[5,6]. Previous studies suggested that PM2.5 has potential immunotoxicity and could be a risk factor for infection[7–9]. Various studies have found a positive association between PM2.5 exposure and risk for respiratory infections[10–13]. However, only limited studies have investigated the effect of PM2.5 on non-respiratory infections and there were

conflicting results[14–17]. Due to the lack of consistent evidence, further study is required in this field.

PM2.5 is a complex mixture of multiple constituents that come from different sources[18]. Different chemical constituents likely have different immunotoxic profiles which could lead to different health impacts. Identifying the constituent-specific effects could help us understand the underlying pathways by which different components and sources of PM2.5 lead to adverse health outcomes and provide information for targeted interventions.

In this study, we investigated the associations between long-term exposure to PM2.5 and hospital admissions from non-respiratory infections among older adults in the US and identified the constituents and source specific PM2.5 which contribute most to the adverse health effects.

Here, we show that ZIP codes with higher PM2.5 concentration had higher rates of hospital admissions from non-respiratory

[1]Department of Epidemiology, Harvard T.H. Chan School of Public Health, Boston, MA, USA. [2]Department of Environmental Health, Harvard T.H. Chan School of Public Health, Boston, MA, USA. [3]Department of Biostatistics, Harvard T.H. Chan School of Public Health, Boston, MA, USA. ✉e-mail: yfeng@g.harvard.edu

infections. Across the 15 PM2.5 constituents we examined, sulfates, Nickel and Copper contributed most to the observed association between PM2.5 and the outcome. PM2.5 sourced from oil combustion, coal burning, traffic, dirt, and regionally transported nitrates had the strongest associations with admissions from different non-respiratory infection.

## Results

### ZIP code-level characteristics

Hospital admissions of non-respiratory infections from 39,296 ZIP codes between 2000 and 2016 were included in the analysis. In total, 13,724,218 admissions from non-respiratory infections were identified. This study included data from 67,005,279 Medicare beneficiaries, among whom 37,037,978 (55.3%) were female, 56,700,531 (84.6%) were White, 5,788,605 (8.6%) were Black and 11,837,647 (17.6%) were ever eligible for Medicaid. Among the 436,577 included ZIP code-years, the median number of beneficiaries was 536, while the median percentage of beneficiaries younger than 75 years-old was 55.2%, the median percentage of female beneficiaries was 55.6%, median percentage of White beneficiaries was 95.9%. Other ZIP code-level characteristics controlled for in the analysis, including percentage of smokers, SES, and meteorology are summarized in Table 1.

### PM2.5 and non-respiratory infection

Across all the included ZIP code-years, the median level of PM2.5 was 9.7 $\mu g\, m^{-3}$ (IQR 7.7–11.8). The median admission rate of non-respiratory infections was 25.1 (IQR 16.3–35.2) per 1000 person-years. The distribution of PM2.5 levels and rate of admission from non-respiratory infections in 2008 are shown in Figs. 1 and 2. After adjusting for the covariates, each standard deviation (3.7-$\mu g\, m^{-3}$) increase in PM2.5 was associated with a 10.8% (95%CI 10.8–11.2%) increase in the admission rate from non-respiratory infection. PM2.5 was also significantly associated with increased admission rates for three sub-types of non-respiratory infection. The increases in admission rate associated with each standard deviation increase in PM2.5 were 6.8% (95%CI, 6.4–7.2%), 12.0% (95%CI, 11.6–12.0%) and 12.8% (95%CI, 12.4–13.2%) for intestinal infections, urinary tract infections

and septicemia respectively (Table 2). When we restricted the analysis to ZIP code-years with PM2.5 ≤ 9 $\mu g\, m^{-3}$, a standard deviation increase in PM2.5 was associated with 21.5% (95%CI 20.6–22.8%) increase in rate of hospital admissions from non-respiratory infections (Table 2).

### Mixture effect of PM2.5 constituents

Each decile increase in the mixture was associated with a 10.3% (95%CI 10.2–10.4%) increase in admission rate of non-respiratory infection, while the constituents with the highest relative importance were $SO_4$ (30.8%), Ni (22.5%) and Cu (15.3%) (Fig. 3). Similarly, each decile increase in the mixture was associated with 10.2% (95%CI 10.1–10.4%), 12.7% (95%CI 12.4–13.0%), and 12.0% (95%CI 11.8–12.3%) increases in admission rate for septicemia, intestinal infections and urinary tract infections, respectively. The constituents with the highest relative importance were Ni (19.7%), $SO_4$ (17.4%) and Cu (15.6%) for septicemia; Cu (37.3%), Ni (20.0%) and $SO_4$ (18.4%) for intestinal infections and $SO_4$ (34.2%), Ni (14.2%), Fe (13.5%) for urinary tract infections. Each decile increase in the mixture was associated with 4.2% (95%CI 3.2–5.2%) increase in admission rate of CNS infections while the constituents with highest relative importance were Si (25.2%), OC (16.2%) and Ca (14.4%). Detailed relative importance of constituents for the four sub-types of non-respiratory infections are shown in Supplementary Fig. S1–4. All the results were adjusted for covariates mentioned above.

### Source specific PM2.5 and non-respiratory infections

Geographical distributions for the clusters within which we conducted separate NMF are shown in Supplementary Fig. S5–7. The major sources of PM2.5 identified from our constituent data were coal burning, traffic, oil combustion, soil, biomass burning and regionally transported nitrates. Coal burning was identified in all of the nine strata while oil combustion was identified in eight strata; soil and traffic were identified in seven strata. Detailed results from NMF are shown in supplementary material (Figs. S8–S16).

After summarizing over all the strata, PM2.5 from five out of the six identified sources was significantly associated with increased admission rate from non-respiratory infections. The estimated increases in admission rate from non-respiratory infections for each one standard deviation increase in source specific PM2.5 were 14.5% (95%CI 7.6–21.8%), 18.2% (95%CI 7.2–30.2%), 20.6% (95%CI 5.6–37.9%), 8.9% (95%CI 0.3–18.4%), and 7.8% (95%CI 0.6–15.5%) for oil combustion, coal burning, traffic, soil and regionally transported nitrate respectively. PM2.5 from oil combustion and coal burning were associated with increased hospital admissions from intestinal infections, urinary tract infections and septicemia; while PM2.5 from traffic was associated with increased admission from all four subtypes of non-respiratory infections. Detailed effect estimates of each of the source-specific PM2.5 are shown in Table 3.

## Discussion

In this study among older adults across the contiguous US, we observed that ZIP codes with higher PM2.5 concentrations had increased admission rates from total and different subtypes of non-respiratory infections including intestinal infections, urinary tract infections and septicemia. The associations were observed even at lower levels of PM2.5 including levels below the US EPA's recently proposed standards, after adjusting for multiple confounders. When we examined PM2.5 constituents, $SO_4$, Ni and Cu contributed most to the association between the constituent mixture and hospital admissions from non-respiratory infections. Based on our source apportionment analysis, PM2.5 sourced from oil combustion, coal burning and traffic had the strongest associations with admissions from different non-respiratory infections, which is generally consistent with the loadings in the mixture analyses, since Ni is a tracer for oil combustion, $SO_4$ for coal combustion, and Cu for traffic particles.

**Table 1 | Characteristics of the included ZIP codes between 2000 and 2016**

|  | Overall |
|---|---|
| ZIP code-years | 436,577 |
| Number of beneficiaries per ZIP code | 536 [222, 1530] |
| Characteristics of Medicare beneficiaries (median percentage [IQR]) | |
| Female | 55.59 [52.55, 58.37] |
| Aged between 65 and 74 | 55.24 [50.33, 59.88] |
| Aged between 75 and 84 | 32.07 [28.83, 35.30] |
| White | 95.86 [86.13, 98.60] |
| Black | 0.90 [0.00, 5.81] |
| Eligible for Medicaid | 10.03 [5.93, 17.26] |
| ZIP code-level contextual characteristics (median [IQR]) | |
| Hispanic (%) | 2.87 [1.04, 8.84] |
| Poverty (%) | 8.46 [5.21, 13.27] |
| Education below high-school (%) | 25.62 [16.06, 37.69] |
| Median Household income ($) | 44575 [35537, 57417] |
| Smoker (%) | 46.76 [42.22, 51.47] |
| Percent beneficiaries had ambulatory visit (%) | 80.35 [76.65, 83.22] |
| Distance to the nearest hospital (KM) | 8.73 [3.04, 17.09] |
| Population density (person/$KM^2$) | 127.38 [35.87, 1110.00] |
| Winter maximal daily temperature (°C) | 6.88 [2.17, 13.37] |
| Summer maximal daily temperature (°C) | 29.51 [27.04, 32.25] |

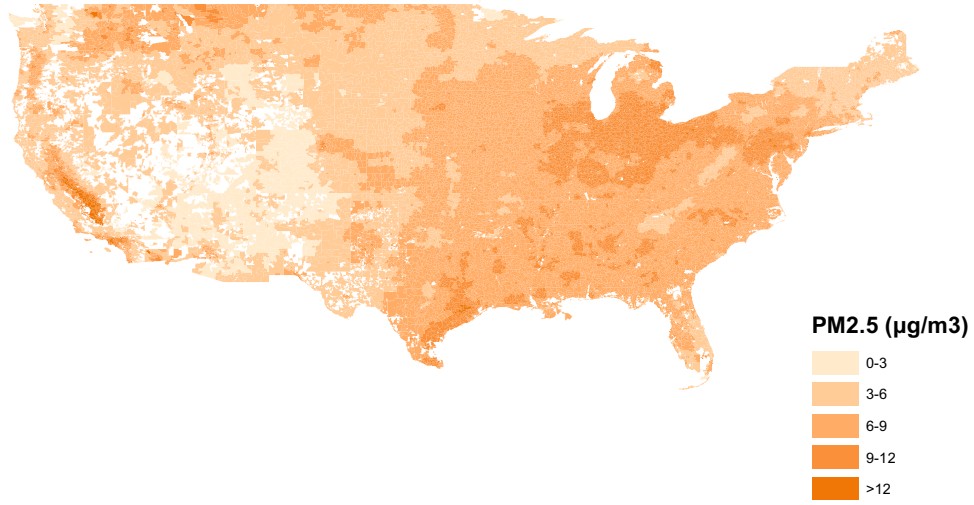

**Fig. 1 | ZIP code-level PM2.5 in the US in 2008.** Source data are provided as a Source Data file. The figure was created from ArcMap 10.7.

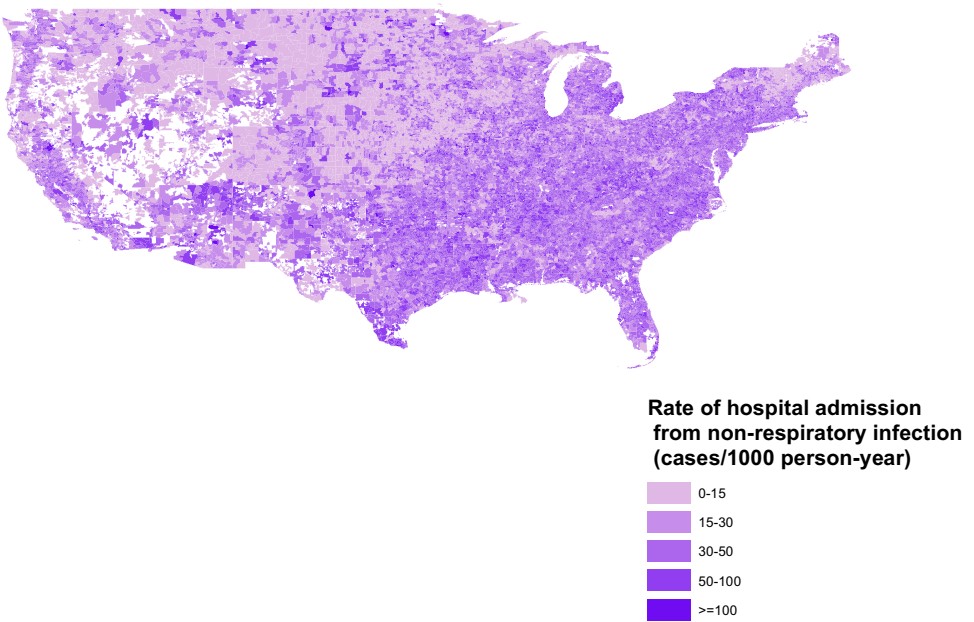

**Fig. 2 | Rate of hospital admissions from non-respiratory infections in the US in 2008.** The rates of hospital admissions are indicated by the Source data are provided as a Source Data file. The figure was created from ArcMap 10.7.

Positive associations between PM2.5 and infections have been found in many studies, most of which focused on respiratory infections such as pneumonia and bronchiolitis[6,13,19–21]. However, associations between PM2.5 and non-respiratory infections have also been observed. A previous study among the US Medicare population found that each $1 \mu g \, m^{-3}$ increase in short-term PM2.5 was associated with 0.41%, 0.39%, 0.13% and 0.09% increases in hospital admission rates from septicemia, urinary tract infections, skin and subcutaneous tissue infections and intestinal infections[20]. However, that study examined short-term exposures to PM2.5 and not long-term exposures as we have done. A study from China observed that each $1 \mu g \, m^{-3}$ increase in short-term PM2.5 was associated with a 0.93% increase in rate of hospital admissions from bacterial infections of unspecified site and a 0.97% increase in the rate of hospital admissions from intestinal infections[17]. Our study further supports that PM2.5 is a risk factor for non-respiratory infections and is the first we know of to examine long-term exposures to particles, particle components, and particle sources. The potential pathways through which PM2.5 affects non-

respiratory infections have not been thoroughly studied yet. An animal study suggested that exposure to PM2.5 down-regulated IL-1β and IFN-β, which led to increased susceptibility to viral infections[22]. Moreover, chronic exposure to PM2.5 could alter phagocytic activity and superoxide dismutase (SOD) activity, which affects immune response to pathogens[23]. The immunotoxicity potentially plays a key role in the observed association between PM2.5 and non-respiratory infections.

We observed that sulfate, Ni and Cu had the largest contribution to the effect of PM2.5 mixture on hospital admissions from non-respiratory infections. Ni is a heavy metal with high immunotoxicity. Previous studies suggested Ni exposure could increase production of reactive oxygen species, reduce the activity of SOD and catalase, and induce mitochondrial dysfunction, which are all interrelated with the function of the immune system[24]. Airborne Ni is predominantly from the combustion of heavy fuel oil, but other sources include metallurgy, stainless steel production and other industrial sources. An animal study found that the soluble Ni component could alter the immune

defense in rats and increase their vulnerability to bacterial infections[25]. Similar to Ni, Cu is also a divalent cation that could affect the immune system and oxidative stress, including through Fenton chemistry. Exposure to high levels of Cu increases the production of pro-inflammatory cytokines and ROS[26]. An in vitro study found that copper

## Table 2 | Association between PM2.5 and rate of hospital admissions from non-respiratory disease among Medicare beneficiaries between 2000 and 2016

| Outcome | Admission rate ratio (95%CI) | |
|---|---|---|
| | All ZIP code-years | ZIP code-years with PM2.5 ≤ 9 μg m⁻³ |
| Non-respiratory infection | 1.108 (1.108–1.112) | 1.215 (1.206–1.228) |
| CNS infection | 1.007 (0.996–1.022) | 0.949 (0.921–0.982) |
| Intestinal infection | 1.068 (1.064–1.072) | 1.189 (1.173–1.206) |
| Urinary tract infection | 1.120 (1.116–1.12) | 1.236 (1.223–1.245) |
| Septicemia | 1.128 (1.124–1.132) | 1.228 (1.215–1.241) |

Admission rate ratio was calculated for each standard deviation (3.7 μg m⁻³) increase in PM2.5. All the associations were adjusted for percentage of beneficiaries who were female, percentage of beneficiaries who were aged between 65 and 74, percentage of beneficiaries who were aged between 75 and 84, percentage of beneficiaries who were White, percent of beneficiaries who were Black, percentage of beneficiaries who were eligible for Medicaid, percentage of population living in poverty, percentage of population having education less than high school, percent smokers, median household income, percentage of population who were on public assistance, average annual percent of Medicare enrollees having at least one ambulatory visit to a primary care clinician, distance to nearest hospital, population density, summer maximal daily temperature, winter maximal daily temperature at ZIP code-level and calendar year.

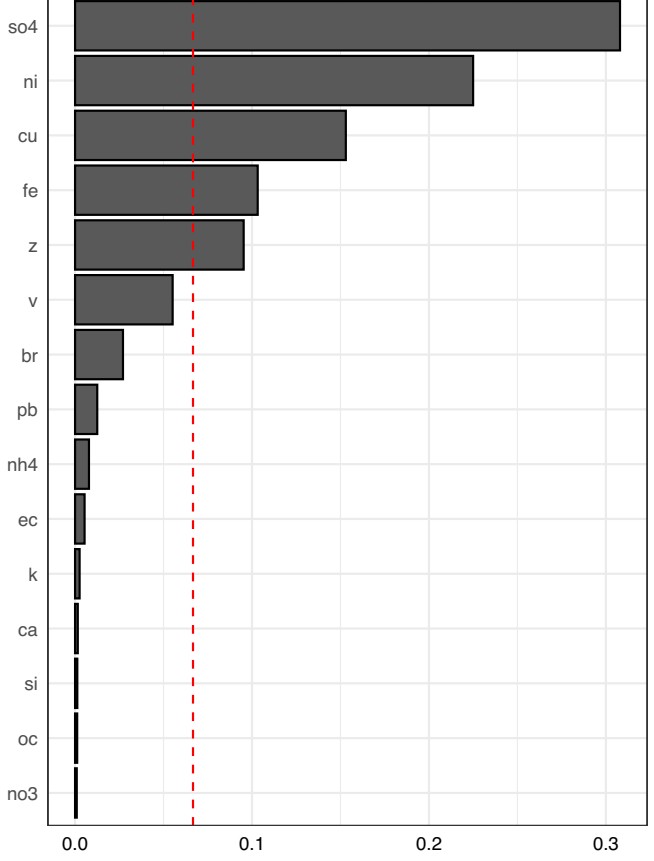

**Fig. 3 | Weights of PM2.5 constituents in the mixture index for the association between the particle mixture and non-respiratory infections.** The dash line indicated the suggested threshold for the most influential constituents. Source data are provided as a Source Data file.

could induce the apoptosis of monocytes. Airborne Cu is primarily a non-tailpipe traffic emission, primarily from brake wear. Other sources include industry and smelters. Sulfate particles derive primarily from coal combustion and have been observed to have the strongest association with mortality across different PM2.5 constituents in a number of studies[27,28]. However, evidence on the association between sulfate and infections is still lacking. A recent study observed that exposure to sulfate particles was associated with increased length of stay from hand-foot-mouth disease[29]. One potential mechanism for the observed effect could be that acidic sulfates turn the metal oxides from PM2.5 into metal ions which are soluble in the lung lining fluid, producing oxidative compounds and ultimately perturb the immune system[30–32]. It is plausible that sulfate interplays with metals such as nickel and copper and leads to the observed association between these PM2.5 constituents and non-respiratory infections.

PM2.5 sourced from coal burning, oil combustion, and traffic was identified in most of the strata and significantly associated with increased admission rate from total non-respiratory infections and most subtypes. These results are in accordance with our mixture analysis, given that sulfates, Nickel and Copper are the main constituents of PM2.5 sourced from coal burning, oil combustion and non-exhaust traffic. It was estimated that more than 80% of the PM2.5 in the US was from fossil fuel combustion including coal, petrol, and diesel and 0.36 million excess deaths in the country during 2012 were attributable to these sources[33]. Previous epidemiological studies suggested that, when compared to natural sources, PM2.5 from fossil fuel combustion has a greater adverse effect on mortality[27,28,34]. Thurston et al observed that coal combustion PM2.5 was associated with increased mortality from ischemic heart disease and that it showed a larger effect than overall PM2.5 in general[28]. Moreover, PM2.5 sourced from fossil fuel burning has also been found to be associated with multiple diseases including pneumonia, psychiatric disorders, and cancer[33,35,36]. Road traffic is another major contributor to the PM2.5 and has been found to be associated with many adverse health outcomes[37–39]. Traffic related PM can be categorized as exhaust and non-exhaust sources. Exhaust particles mainly come from tailpipe emissions while non-exhaust particles mainly come from the wear of tires and brakes, and also the abrasion of road surface[40]. A recent meta-analysis suggested that air pollution sourced from traffic emissions were associated with multiple adverse health outcomes including low birthweight, asthma onset, cardiovascular events, and mortality from multiple causes[41]. The results from our study provide further evidence for the harmful effect of PM2.5 sourced from fossil fuel and traffic on health, suggesting that regulations on PM2.5 could potentially focus on these sources.

The observed association between PM2.5 and its constituents and the outcomes were consistent across different subtypes of non-respiratory infections. However, the association between PM2.5 and CNS infection was much weaker when compared to other subtypes. One plausible explanation would be that the blood-brain barrier provides unique protection to the CNS and thus may be less affected by the environmental risk factors such as air pollution[42]. Related evidence is still sparse and, therefore, more studies are needed in order to elucidate the observed difference between CNS infections and other subtypes of non-respiratory infections.

This study has several strengths. First, this is the first study to comprehensively evaluate the impact of PM2.5 on non-respiratory infections. Second, using the data of PM2.5 constituents across the contiguous US, the study was able to identify the constituents and source-specific PM2.5 which had the strongest associations with different non-respiratory infection outcomes. These results could potentially help elucidate the potential mechanisms behind the immunotoxicity of PM2.5 and its constituents. Moreover, the results from mixture and source specific analysis are informative for targeted prevention strategies and policy making. Third, this study included more than 60 million participants, and most of the Medicare fee-for-

**Table 3 | Association between source-specific PM2.5 and rate of hospital admissions from non-respiratory infections among Medicare beneficiaries between 2000 and 2016**

| Admission Rate Ratio | | | | | |
|---|---|---|---|---|---|
| | **Non-respiratory infection** | **CNS infection** | **Intestinal infection** | **Urinary tract infection** | **Septicemia** |
| Oil combustion | 1.145 (1.076–1.218) | 1.004 (0.988–1.02) | 1.171 (1.085–1.264) | 1.148 (1.051–1.255) | 1.135 (1.074–1.199) |
| Soil, dirt | 1.089 (1.003–1.184) | 1.112 (1.065–1.162) | 1.105 (0.962–1.27) | 1.153 (0.99–1.343) | 1.032 (0.99–1.075) |
| Coal burning | 1.182 (1.072–1.302) | 1.007 (0.988–1.026) | 1.161 (1.013–1.332) | 1.226 (1.076–1.397) | 1.154 (1.068–1.248) |
| Traffic | 1.206 (1.056–1.379) | 1.056 (1.011–1.103) | 1.239 (1.038–1.479) | 1.278 (1.042–1.567) | 1.187 (1.067–1.321) |
| Biomass | 1.089 (0.996–1.192) | 1.059 (1.012–1.109) | 1.091 (0.966–1.234) | 1.088 (0.942–1.256) | 1.135 (1.075–1.199) |
| Regionally transported nitrate | 1.078 (1.006–1.155) | 0.987 (0.943–1.034) | 0.992 (0.981–1.003) | 1.051 (1.017–1.085) | 1.112 (1–1.236) |

Admission rate ratio was calculated for each standard deviation increase in source-specific PM2.5. All the associations were adjusted for percentage of beneficiaries who were female, percentage of beneficiaries who were aged above between 65 and 74, percentage of beneficiaries who were aged between 75 and 84, percentage of beneficiaries who were White, percent of beneficiaries who were Black, percentage of beneficiaries who were eligible for Medicaid, percentage of population living in poverty, percentage of population had education less than high school, percent smoker, median household income, percentage of population who were on public assistance, average annual percent of Medicare enrollees having at least one ambulatory visit to a primary care clinician, distance to nearest hospital, population density, summer maximal daily temperature, winter maximal daily temperature at ZIP code-level and calendar year.

service beneficiaries between 2000 and 2016 in the analysis, which is a representative sample of the older adults across the US, therefore, the results are likely to be generalizable to older population. The large sample size also provided enough power for us to evaluate the association between the exposures and multiple subtypes of non-respiratory infections.

The study also has limitations. First, the exposure data in this study are at ZIP code level instead of individual level. The participants' individual exposure level within a same ZIP code could vary due to different factors. The exposure data were aggregated from grid cell level to ZIP code level. Therefore, the measurement error of the exposure could be affected by the size of ZIP code area and population, which is potentially non-differential. However, the within ZIP code coefficients of variation of the exposure data are small for the major PM2.5 components, suggesting that the potential measurement error arising from the aggregation was small. Moreover, previous studies suggested that ZIP code level air pollution is a valid proxy for individual level exposure and could avoid some of the impact of personal level confounding[43]. Second, the ZIP code level total PM2.5 and PM2.5 constituents in this study are not weighted by population. However, population-weighted exposure estimates could potentially lead to larger measurement error and further bias the result estimates. Third, the outcomes of this study were hospital admissions, which only captures the cases that were hospitalized or the most severe non-respiratory infection cases. It is unclear whether these results also apply to the milder cases. Further study which captures the outpatient cases are needed in the field. Fourth, this study was limited to Medicare beneficiaries who were 65 or older. Considering that older adults are more vulnerable to the adverse effect of air pollution, the results from this study might not be generalized to younger population. Moreover, since the metal components had substantial weights in contributing to the weighted quantile sum, but very small mass concentrations, the effect size estimate per weighted decile of the mixture cannot be directly compared to the effects per unit mass of PM2.5. However, these small mass components also contributed importantly to the source apportionment study, which may partially explain the larger effect size for sources, while this may also reflect the unimportance of other components of PM2.5 that were not included in our component mixture. Lastly, we are only able to obtain limited data on individual level confounders in this study. To reduce residual confounding, confounders such as smoking and SES were controlled at the contextual level.

In conclusion, higher ZIP code level PM2.5 exposure was associated with increased rate of hospital admissions from non-respiratory infections and the association remained robust even in areas with lower PM2.5, continuing below PM2.5 concentrations of 9 µg m$^{-3}$.

Sulfates, Nickel, and Copper play the most important role in the effect of the PM2.5 mixture on non-respiratory infections. PM2.5 sourced from fuel oil combustion, coal burning and traffic had larger effects on admission from non-respiratory infections when compared to PM2.5 from other sources. PM2.5 effects on non-respiratory infections has been understudied and this study provides evidence that this pathway could be an important additional impact from PM2.5 and should be considered in evaluating the adequacy of current PM2.5 standards.

## Methods
### Study population
We included all beneficiaries who enrolled in Medicare fee for service and were aged 65 and older between 2000 and 2016. Medicare is a national health insurance program in the U.S. which provides coverage mainly for those who are 65 years of age or older[44]. Beneficiaries entered the open cohort on January 1st 2000 or the first January 1st after their enrollment, which ever came later, and were followed until the date of death, or December 31st, 2016, whichever came earlier. This study is approved by the IRB of the Harvard T.H. Chan School of Public Health. Informed consent was waived because this study conducted secondary analysis of deidentified data.

### Outcome
The outcomes of this study were hospital admissions from non-respiratory infections, and its subtypes including central nervous system (CNS) infections, intestinal infections, urinary tract infections and septicemia. Data on hospital admissions including ICD (International Classification of Diseases) Diagnosis codes at discharge and date of admissions among Medicare beneficiaries between 2000-2016 were extracted from the Medicare Provider Analysis and Review (MEDPAR) data file. Hospital admission data could include multiple diagnosis codes for each admission record. In this study, we defined the outcome as having a principal discharge diagnosis code of non-respiratory infections.

For hospital admission records with ICD-9 diagnosis codes, we applied the categorizing scheme from the Clinical Classification Software for ICD-9-CM (CCS) to categorize over 14,000 ICD-9 diagnosis codes into 280 clinically meaningful and mutually exclusive diagnosis groups[45]. For hospital admission records with ICD-10 diagnosis code, we applied the categorizing scheme from the Clinical Classification Software Refined for ICD-10-CM (CCSR) to categorize over 70,000 ICD-10 diagnosis codes into 530 clinically meaningful diagnosis groups[46]. Based on CCS and CCSR, ICD diagnosis from hospital admission records were categorized into total non-respiratory infections, CNS infections, intestinal infections, urinary tract infections and septicemia.

For admission records with ICD-9 diagnosis codes, non-respiratory infection was defined as diagnosis code being within one of group 2–9, 76–78, 90, 135, 159, 197, 201 and 248 of CCS scheme; CNS infection was defined as diagnosis code being within one of group 76–78; intestinal infection was defined as diagnosis code being within group 135; urinary tract infection was defined as diagnosis code being within group 159; septicemia was defined as diagnosis code being within group 2.

For CCSR scheme, each ICD-10 diagnosis code could exist in more than one diagnosis group. Therefore, for admission records with ICD-10 diagnosis codes, non-respiratory infection was defined as being in group INF002-004, INF006-011, MUS001-002, MUS027, NVS001-NVS003, GEN001, SKN001 and DIG001 but not in group RSP002-006. CNS infection was defined as diagnosis code being within either one of group NVS001-003; intestinal infection was defined as diagnosis code being in group DIG001; urinary tract infection was defined as diagnosis code being in group GEN001; septicemia was defined as diagnosis group being in group INF002.

The number of hospital admissions from non-respiratory infections and its subtypes were calculated for each year within each ZIP code.

## Environmental exposure data
Levels of ambient PM2.5 and its constituents across the contiguous US between 2000 and 2016 were estimated from validated ensemble machine learning models developed by our grouped previously. Details of the exposure modeling are described elsewhere[47–49]. Daily PM2.5 at a 1 km*1 km grid cell level was estimated by combining predictions from random forest, gradient boosting, and neural network models in a geographically weighted regression ensemble. Predictors of the model for PM2.5 included aerosol optical depth, meteorology data, chemical transport model simulations and land-use data[47]. The cross-validated (CV) $R^2$ of the annual PM2.5 prediction model was 0.89.

Annual mean levels of 15 PM2.5 constituents [elemental carbon (EC), ammonium ($NH_4^+$), nitrate ($NO_3^-$), organic carbon (OC), and sulfate ($SO_4^{2-}$), bromine (Br), calcium (Ca), copper (Cu), iron (Fe), potassium (K), nickel (Ni), lead (Pb), silicon (Si), vanadium (V), and zinc (Zn)] were estimated at a 50 m*50 m grid cell level in urban areas and at a 1 km*1 km level in rural areas. The estimates incorporated predictions from multiple machine-learning models including random forest (RF), stochastic gradient boosting (GBM), extreme gradient boosting (XGB), cubist, and K-nearest neighbors (KNN) models. These were ensembled using a support vector machine (SVM). Like the prediction models for PM2.5, models for PM2.5 constituents included a large number of predictors which included satellite observations, meteorology data and novel land use covariates[48,49]. The CV $R^2$ ranged from 0.80 to 0.96 across different constituents. The constituents predicted comprise most, but not all of the total mass of PM2.5.

Grid cell level PM2.5 and constituent concentrations were then aggregated to the ZIP code level based on previously described methods[50]. The coefficient of variation for each of the exposure within ZIP codes are provided in supplementary table 1.

## Covariates
We obtained demographic information (age, sex, race), ZIP code of residence and Medicaid eligibility of the Medicare beneficiaries from the Medicare denominator file. Sex information in Medicare data was obtained from multiple sources including self-reported at enrollment, claims data or Electronic Health Records. Information on beneficiaries' ZIP code, age, and Medicare eligibility were updated annually. ZIP code level socioeconomic status (SES) data including percentage of Hispanics, percentage of population that had less than high school education, median household income, percentage of population who were on public assistance and percentage of population living in poverty

were directly obtained from American Community Survey 5-year estimates between 2011 and 2016[51] or linearly interpolated from the 2000 and 2010 US Decennial Census[52]. Data on the percentage of smokers within a ZIP code was obtained from Behavioral Risk Factor Surveillance System. Annual percent of Medicare enrollees having at least one ambulatory visit to a primary care clinician and distance to nearest hospital at ZIP code level were obtained or calculated from data from the Dartmouth Atlas of Healthcare website[53]. ZIP code-level maximal daily temperature in the summer and maximal daily temperature in the winter was calculated from the Gridded Surface Meteorological (gridMET) Dataset[54]. To account for potential temporal trends, we also included indicatory variables of calendar year as covariates.

## Statistical analysis
Individual level data of Medicare beneficiaries were first aggregated to counts of events by ZIP code and year and merged with the ZIP code-level covariates. Within each ZIP code-year stratum, we calculated the total counts of non-respiratory infections as well as its subtypes, total number of Medicare beneficiaries, percent of female beneficiaries, percent of Black beneficiaries, and percent of beneficiaries who were also eligible for Medicaid. In our analysis, we only included ZIP codes which had more than 100 beneficiaries.

Associations between PM2.5 and the admission rates of our five outcomes, namely total non-respiratory infections, central nervous system (CNS) infections, intestinal infections, urinary tract infections, and septicemia were investigated using multivariable quasi-Poisson regression models below.

$$\log\left(admissioncount_{ijk}\right) = \beta_0 + \beta_1 PM2.5 + \beta' Z + \log(beneficiarycount_{ij}) + \varepsilon_{ijk}$$

Where $i$ indicates the ith ZIP code, $j$ indicates the jth year, $k$ indicates the kth outcome (total non-respiratory infection and its subtypes), $Z$ indicates the matrix of covariates mentioned above. Admission counts indicates the number of hospital admissions for the corresponding outcome while beneficiary count indicates the number of Medicare beneficiaries aged 65 or older who were alive on January 1st of the corresponding year within the ZIP code. Quasi-Poisson regression was used because the outcome of the study was counts data and the variance of the outcome was larger than the mean of the outcome.

The US Environmental Protection Agency is considering lowering the standard of annual PM2.5 to 9–10 μg m⁻³. To evaluate the effect of PM2.5 at levels below the proposed standards, we conducted the same analysis restricted to ZIP code-years where PM2.5 ≤ 9 μg m⁻³.

Given that PM2.5 is a mixture of multiple constituents, within which correlations exist, we used weighted quantile sum regression to investigate the mixture effects of the different constituents. Weighted quantile sum regression is a modeling technique which can identify the association between mixtures and the outcome of interest while reducing the impact of high collinearity. A detailed description of the weighted quantile sum method is provided in the supplementary information. Briefly, weighted quantile sum regression estimates the relative contribution of each constituent and generates a mixture index as a linear combination of different constituents. It assumes each quantile increase in the mixture index is linearly and unidirectionally associated with the outcome[55]. Using weighted quantile sum regression with a quasi-Poisson link, we estimated the association between each one decile increase in the PM2.5 mixture and our five outcomes of interest while estimating the relative contribution from each of the components within the PM2.5 mixture. The weighted quantile sum regression was run with 100 bootstrap samples and all the weights were constrained to be positive.

To identify source specific effects of PM2.5, we used non-negative matrix factorization (NMF) to conduct source apportionment on our ZIP code-level PM2.5 constituent data. Similar to principal component

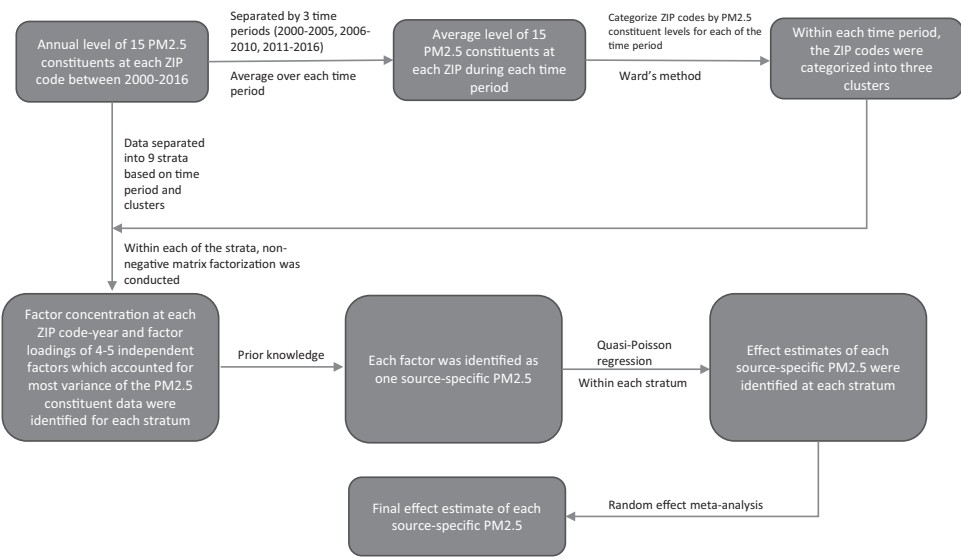

**Fig. 4 | Flowchart for identify source specific effects of PM2.5.** The 15 PM2.5 constituents included Br, Ca, Cu, EC, Fe, K, NH4, Ni, NO3, OC, Pb, Si, SO4, V and Zn.

analysis, NMF is a method for dimensionality reduction which explains the observed multidimensional data using limited number of bases[56]. However, NMF constrains the matrix components and the mixture coefficients to non-negative values, which is more appropriate for mass concentrations[57]. The regulations for the sources of air pollution varied differently over time, and hence the proportion of each component's contribution to each source was likely to also vary over time. Therefore, we conducted source apportionment separately by time periods. To ensure enough sample size for NMF, we separated our data by three time periods (2000–2005, 2006–2010 and 2011–2016). The contribution of different sources of PM2.5 also vary across regions. Therefore, within each time period, we summarized the ZIP code mean concentration for each constituent. Using Ward's hierarchical clustering, the contiguous US was categorized into three clusters with similar patterns of PM2.5 constituents for each of the time periods. Within each of the nine stratum (3 clusters * 3 time periods), NMF was conducted to identify 4–5 independent factors which accounted for most of variance of the PM2.5 constituent data and the source factors levels for each observation[35]. Based on the factor loadings of the constituents and our prior knowledge on the trace elements of PM2.5, we identified the source for each factor[58] (details are provided in the supplementary information). The units of source factor levels were first converted to $\mu g\, m^{-3}$ and rescaled by the standard deviation of each source factor (see supplementary methods). Within each stratum, associations between source specific factors and non-respiratory infection outcomes were estimated using quasi-Poisson regression models, and all the source factors were mutually adjusted. The total effect for each source was estimated by combining the stratum specific effects using random effect meta-analysis (flowchart is shown in Fig. 4). Details of the random effect meta-analysis are provided in the supplementary information.

All the analyses were adjusted for the confounders listed in the covariate section. To account for the autocorrelations within ZIP codes across years, we incorporated robust standard error in all of the effect estimates. Statistical analyses were conducted using R 4.1.3[59]. The analytic code for this project is available in Supplementary Code published alongside with this manuscript.

### Reporting summary
Further information on research design is available in the Nature Portfolio Reporting Summary linked to this article.

## Data availability
The data of the Medicare beneficiaries are available under restricted access due to the requirements from the Center for Medicare and Medicaid Services (CMS). Researchers can submit their data request to CMS and the request will be forwarded to the CMS dissemination contractor for processing. Processing of the data takes approximately 2–4 weeks (depending on the number and years of files being requested). However, the data use agreement prevents us from sharing that data and so are not publicly available. According to the DUA, our group could only access the data on the level 3 cluster of Harvard University with controlled access and cannot download the data. The air pollution data used in this study are publicly available on the SEDAC website: https://sedac.ciesin.columbia.edu/data/set/aqdh-pm2-5-component-ec-nh4-no3-oc-so4-50m-1km-contiguous-us-2000-2019 and https://sedac.ciesin.columbia.edu/data/set/aqdh-pm2-5-component-trace-elements-50m-1km-contiguous-us-2000-2019 and https://sedac.ciesin.columbia.edu/data/set/aqdh-pm2-5-o3-no2-concentrations-zipcode-contiguous-us-2000-2016. The data from American Community Survey and US census are available at https://data.census.gov/. Data from BFRSS are available at https://www.cdc.gov/brfss/annual_data/annual_data.htm. Data from Dartmouth Healthcare Atlas are available at https://data.dartmouthatlas.org/. Source data of the figures are provided with this paper. Source data are provided with this paper.

## Code availability
The analytic codes of this study are publicly available from https://github.com/yatkan/PM2.5_nonrespiratoryInfection_NC/tree/main. Code is also available in Supplementary Code published alongside this manuscript.

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

## Acknowledgements

This study was funded by JS's National Institute of Environmental Health Science (NIEHS) grant RO1 ESO23418 and NIEHS P30-000002.

## Author contributions

YF contributed to conceptualization, methodology, data analysis and writing original draft. EC contributed to data compiling, methodology and manuscript editing & reviewing. YW, XQ and TJ contributed to methodology and manuscript editing & reviewing. FD contributed to manuscript editing & reviewing. JS contributed to funding, conceptualization, methodology and manuscript editing & reviewing.

## Competing interests

J.S. has been an expert witness for the United States Department of Justice on cases involving violations of the Clean Air Act. The other authors declare no competing interests.
