## [Peer Review File · Nature Communications]

Long-term Exposure to Ambient PM_{2.5}, Particulate Constituents and Hospital Admissions from Non-respiratory InfectionREVIEWER COMMENTS

Reviewer #1 (Remarks to the Author):

General: This is a generally excellent article evaluating a novel set of health outcomes for their relationship to PM2.5 mixtures and sources. There are a few places where clarifications are needed, and some additional discussion of potential uncertainties introduced through the use of zip code level exposure surrogates is warranted. The findings that non-respiratory infections are significantly and substantially affected by exposure to PM2.5 mixtures and sources at levels that are at or below the current standards adds additional weight to the health burdens that U.S. residents face from this pollutant. The work builds on other recent studies using the Medicare cohort data, and uses state of the art estimates of PM2.5 and components, although the use of zip codes as the unit of study raises some concerns. While the work follows previously published uses of the Medicare cohort to evaluate air pollution impacts, there are acknowledged limitations due to the inability to control for individual level risk factors. The use of ecological controls for many of those risk factors is appropriate but cannot fully account for the lack of controls for individual level factors. This is not a fatal flaw, and the authors appropriately note these limitations in their discussion of results.

The paper is generally well written and understandable, however, there are a fairly large number of grammatical errors. I have tried to note where these occur, however, I am sure that I did not catch all of them. Please check especially for proper use of plural forms. For example, in most cases it should be “hospital admissions” and “infections”.

Specific comments are provided below.

Title: I think this would be better worded as “Long-term Exposure to Ambient PM2.5, Particulate Constituents and Hospital Admissions from Non-respiratory Infections”

Abstract: Line 15. Wording. Suggest “Using data from Medicare beneficiaries and a high-resolution dataset....”

Line 17: Change “infection” to “infections”

Line 35: Suggest deleting “Therefore, non-respiratory infections are also worth our concern.” It is duplicative of the 3rd sentence in the paragraph and makes it about value judgements.

Line 36-37: While I agree and this may be established fact at this point, it seems appropriate to include a reference here. For example you could cite to the most recent U.S. EPA Integrated Science Assessment for PM2.5. U.S. EPA. Integrated Science Assessment (ISA) for Particulate Matter (Final Report, Dec 2019). U.S. Environmental Protection Agency, Washington, DC, EPA/600/R-19/188, 2019.

<https://cfpub.epa.gov/ncea/isa/recordisplay.cfm?deid=347534>

Line 58. Please clarify. Did the follow up on an individual end with first hospitalization for a non-respiratory infection? What about subsequent infections? PM2.5 could be associated with multiple hospitalizations for infections, or could even be hypothesized to increase repeat hospitalizations for those who had experienced a non-PM2.5 related first admission.

Line 63. See previous comment. If the follow up was only until first admission for non-respiratory infection, then you should clarify that as the outcome, e.g. “The outcomes of this study were first hospital admissions from non-respiratory infections,.....”

Line 109. How are you accounting for the differences in size between zip codes and changes in zip code boundaries between years? Zip codes are not determined by equal population size, nor are they equal in area, so the exposure measurement error from aggregating to the zip code could be an issue as some zip codes could be quite small and others quite large.

Can you provide statistics for the variability in PM2.5 and constituent concentrations within and across zip codes? For example, calculate the coefficient of variation for each zip code, and then provide distributional statistics for the coefficient of variation across zip codes.

This would give a sense of how potential exposure error might differ across zip codes.

Line 137-138. Correct sentence to “To investigate the effect of PM2.5 at lower levels, we conducted the same analysis restricted to ZIP code-years where $PM_{2.5} \leq 9 \mu g/m^3$.”

Line 142. Add “regression” after “Weighted quantile sum”

Line 143. Add “regression” after “Weighted quantile sum”

Line 187. Add “infection” before “admission” and change “admission” to admissions.

Line 195: change to “with $PM_{2.5} \leq 9 \mu g/m^3$, a $1\text{-}\mu g/m^3$ increase in $PM_{2.5}$ was associated....”

Lines 218-236: I think this would be better to just show in table 3, with some high level

summary statements such as in line 223-224 “PM2.5 from oil combustion, coal burning and traffic were 224 associated with increased rate of hospital admission from intestinal infection”. Otherwise, there are a lot of long lists of numbers which are just repeated in table 3.

Lines 239-242: Long run on sentence. Suggest splitting to read “In this study among older adults across the contiguous US, we observed that higher ZIP-code level PM2.5 concentrations were associated with increased admission rates from total and different subtypes of non-respiratory infection including intestinal infection, urinary tract infection and septicemia. The associations were observed even at lower levels of PM2.5.”

Line 243: Awkward phrasing. Suggest rewording to “...and Cu contributed most to the association between....” Or “...and Cu had the highest percentage contribution to the association between...”

Line 244: Change “Based on our sourced apportionment” to “Based on our source apportionment”

Line 245: Change “...and traffic were mostly associated...” to “...and traffic had the strongest associations with....”

Line 266: Unnecessary contraction, reword to “...infection have not...”

Line 271: Reword to “...had the largest contribution to the effect...”

Line 278: Reword to “...in rates and increase vulnerability to....”

Line 279: Reword to “Exposure to high levels of Cu increases the....”

Lines 290-291: Suggest changing to “It is possible” or “It is plausible”. I don’t think that you’ve established a likelihood for the sulfate interactions being the cause of the associations.

Line 297: Something is wrong here. 2.7 million excess deaths in the U.S. would exceed total deaths in 2012. Perhaps you mean 2.7 million deaths globally?

Line 300: Reword to “Thurston et al observed that coal combustion PM2.5 is associated with increased mortality...”

Line 301: Reword to “...and that it showed a larger effect than total PM2.5 (44).”

Line 306: Reword to “Exhaust particles mainly come from tailpipe emissions while non-exhaust particles mainly come....”

Line 316: Reword to “...and thus may be less affected by...”

Line 318: Reword to “study is needed...” The adverb is not needed or justified.

Lines 332-334. I would recommend also noting the potential for the use of zip codes with unequal areas and populations to lead to the potential for differential exposure measurement errors. I don't know how you can definitely state that the measurement error is likely to be non-differential.

Line 347: Reword to "Sulfates, nickel, and copper play the most important role..."

Line 348: Reword to "...had larger effects on.."

Line 350: Reword to "...PM2.5 on non-respiratory infections has been understudied and this study provides evidence that this pathway could be an important additional impact from PM2.5 and should be considered in evaluating the adequacy of current PM2.5 standards."

Reviewer #2 (Remarks to the Author):

The manuscript provides a thorough review of the association between PM2.5 and non-respiratory infections. The authors provide helpful visualizations including maps of the PM2.5 concentrations and the non-respiratory infection rates across the United States. They identify associations but do not overstate the results, as they do not make conclusions about casual relationships. However, the description of the analyses performed is missing important information, making it difficult to understand and reproduce the analyses.

Major revisions

- It is unclear how deaths were treated in the analyses. This should be clarified in the methods section. The sentence, "...were followed until the date of hospitalization from non-respiratory infections, death, or December 31st, 2016, whichever came earlier" suggests a time to event analysis. The analysis section specifies that a quasi-Poisson regression model was used but not how the model accounts for deaths. It is unclear if those who died after they were infected were included in the outcome counts of non-respiratory infections. It is unclear if the researchers were able to identify those who died from an infection without a prior hospitalization and whether these individuals were included in the non-respiratory infection counts.
- A more detailed description of the weighted sum quantile regression would help to clarify how the contribution of constituents were associated with the non-respiratory infection rates. It is unclear how the weighted sum quantile regression was combined with the quasi-

Poisson model to estimate the relative contribution of each constituent on non-respiratory infection. There are no references or models provided for the weighted sum quantile regression and the appendix did not include a detailed discussion of this analysis.

- A more detailed explanation is provided for the source apportionment analysis, but it is spread between the manuscript and an appendix, which makes it difficult for researchers to understand and reproduce the analysis. Important details of the analysis are unclear. It would be helpful for the authors to include an overview of how the cluster analysis, non-negative matrix factorization, quasi-Poisson regression, and the random effect meta-analysis models were combined to obtain the estimates in Table 3. The authors do not provide details about the random effect meta-analysis models in either the manuscript or appendix.

Minor Comments

- It is unclear how the threshold for the most influential constituents was determined. In Figure S1 and S3, we see constituents with weighted concentrations very close to the threshold and they appear to be arbitrarily excluded or included as one of the most influential constituents.
- There are no references provided for the clustering algorithm. This approach is not commonly used so a reference or more detailed explanation is warranted.
- Cardiovascular disease is a major cause of death and not specifically discussed in this manuscript. The authors could provide clarity by including references of studies exploring the relationship between PM2.5 and cardiovascular disease or a sentence on this relationship in the discussion.

Reviewer #3 (Remarks to the Author):

This is an interesting article examining the relationship between hospital admissions for non-respiratory infections and PM2.5 concentrations, and constituent components, using Medicare data from 2000-2016. The data used is impressively comprehensive across in terms of pollutants, hospital admissions, and other covariates. Although examining non-respiratory infections is important and novel, the article has methodological weaknesses that dampen enthusiasm. In particular, concerns over the modeling techniques in dealing

with annual autocorrelations in hospital admissions and pollutants casts doubt on the validity of the results, and therefore what the manuscript is able to add to the extant literature.

Major comments:

The scale of the analysis needs clarification, especially early on in the manuscript, when describing the methods, and when putting the results in to context. For example, the time scale of the exposures is not specified in the abstract (e.g. are these results about daily increases in pollutants, annual increases in pollutants?). When describing the outcome variable in methods, there is good detail about how admissions are classified using ICD codes, but not about the how the infections are counted (e.g., annually? At the zip code level?) until the statistical analysis section. After reading the manuscript, I believe these results indicate that zip codes that experience higher annual averages of PM2.5 also experience higher hospital admissions for non-respiratory infections, but it is not explicitly stated. This comment is largely addressable.

It is not clear how the autocorrelations between pollutant levels in zip codes across years, and similarly the autocorrelations in hospital admissions by zip code across years, is handled. This is a serious concern. The analysis includes indicators for calendar year as covariates, but this does not appear to adequately address the lack of independence in observations from the same zip code across years. This calls the validity of the findings in to question.

Similarly, how do findings handle the problem that the most populated places in the country (and therefore places most likely to have high hospital admissions) also appear to be among the most polluted (as suggested in the Figures)? More directly, it is not clear how much the results are from finding associations between increases in PM2.5 and hospital admissions in a given location, versus comparing hospital admissions between two locations that have different levels of PM2.5?

Grid cell level predicted exposure values for PM2.5 constituents combine predictions from random forests, gradient boosting, and neural networks (paragraph starting page 99). It's

unclear how much sensitivity these predictions may have to the (multitude) of methods used. It's unclear how reliable/good estimates of annual PM2.5 exposure are to actual exposure levels, or how much they might vary given the variety of methods employed to estimate them. How sensitive are exposure estimates of PM2.5 to the (variety) of complex methods employed to estimate them? This is certainly an addressable comment.

Minor comments:

Introduction: Would like more explanation of conflicting extant results of PM2.5 on non-respiratory infections (references 12-15). This could help strengthen the argument for the importance of this manuscript.

The sentence beginning "To investigate the effect of PM2.5 at lower level, we conducted ..." is unclear. Lower level of what? Why is it important to do this?

Statistical analysis should include more support for the use of quasi Poisson models.

The scale of the reporting of hospital admissions increases for changes in PM2.5 concentrations (on the order 3%) is quite different from those for constituents (on the order of 10-20%). It would help to have more context on whether or not these are on similar 'scales,' or what a unit increase or sd increase actually means in terms of pollution.

The sentence "Since the mental components ..." seems to belong in a limitations section, rather than in the first paragraph.

The authors should address how using Medicare data (largely older individuals'), may limit conclusions.

Reviewer #1

Title: I think this would be better worded as “Long-term Exposure to Ambient PM2.5, Particulate Constituents and Hospital Admissions from Non-respiratory Infections”

We have made the suggested change

Abstract: Line 15. Wording. Suggest “Using data from Medicare beneficiaries and a high-resolution dataset....”

Thank you for the suggestion. The sentence has been updated in the abstract.

Line 17: Change “infection” to “infections”

Thank you for the suggestion. This has been corrected in the abstract.

Line 35: Suggest deleting “Therefore, non-respiratory infections are also worth our concern.” It is duplicative of the 3rd sentence in the paragraph and makes it about value judgements.

We agree and have done so.

Line 36-37: While I agree and this may be established fact at this point, it seems appropriate to include a reference here. For example you could cite to the most recent U.S. EPA Integrated Science Assessment for PM2.5. U.S. EPA. Integrated Science Assessment (ISA) for Particulate Matter (Final Report, Dec 2019). U.S. Environmental Protection Agency, Washington, DC, EPA/600/R-19/188, 2019. <https://cfpub.epa.gov/ncea/isa/recordisplay.cfm?deid=347534>

Thank you for the suggestion, the reference has been included in the introduction

Line 58. Please clarify. Did the follow up on an individual end with first hospitalization for a non-respiratory infection? What about subsequent infections? PM2.5 could be associated with multiple hospitalizations for infections, or could even be hypothesized to increase repeat hospitalizations for those who had experienced a non-PM2.5 related first admission.

Thank you for the comments. The patients were followed till death or the end of the study. All the recorded hospital admission from non-respiratory infection are included in this study, not just the first event.

Line 63. See previous comment. If the follow up was only until first admission for non-respiratory infection, then you should clarify that as the outcome, e.g. “The outcomes of this study were first hospital admissions from non-respiratory infections,.....”

Patients were followed till death or administrative censoring. All of the hospital admission from non-respiratory infection during the study period were included. This has been clarified in the Study population under methods section.

Line 109. How are you accounting for the differences in size between zip codes and changes in zip code boundaries between years? Zip codes are not determined by equal population size, nor are they equal in area, so the exposure measurement error from aggregating to the zip code could be an issue as some zip codes could be quite small and others quite large. Can you provide statistics for the variability in PM2.5 and constituent concentrations within and across zip codes? For example, calculate the coefficient of variation for each zip code, and then provide distributional statistics for the coefficient of variation across zip codes. This would give a sense of how potential exposure error might differ across zip codes.

Thank you so much for the suggestions. While aggregating the data to ZIP code level, we apply different boundaries for each of the year. We agree that for larger ZIPs, there is more measurement error because we are aggregating more grid cells, which is a potential limitation of this study. We've included this in the limitation section. We summarized the coefficient of variation of the exposure within each ZIP code across the study period and the results are shown below. This has also been included in the appendix (supplementary table 1). We believe the 5% coefficient of variation within ZIP code indicates that the resultant exposure error is not large, although it is more substantial for some of the trace elements.

Coefficient of variation	
	Median (25th%, 75th%)
PM2.5	5.5% (3.6%, 8.6%)
Br	3.5% (2.5%, 4.9%)
Ca	6.9% (5.3%, 9.2%)
Cu	14.1% (8.9%, 21.6%)
EC	9.6% (6.9%, 13.4%)
Fe	7.0% (5.0%, 10.0%)
K	5.8% (3.9%, 8.7%)
NH4	6.2% (4.8%, 8.4%)
Ni	20.9% (13.6%, 33.6%)
NO3	8.1% (5.6%, 12.0%)
OC	6.3% (4.3%, 9.1%)
Pb	10.3% (7.7%, 13.9%)
Si	6.9% (4.9%, 8.2%)
SO4	3.8% (2.8%, 5.0%)
V	11.9% (8.0%, 18.0%)
Zn	8.4% (6.1%, 11.9%)

Line 137-138. Correct sentence to “To investigate the effect of PM2.5 at lower levels, we conducted the same analysis restricted to ZIP code-years where $PM_{2.5} \leq 9 \mu g/m^3$.”

Thank you for the suggestion, this has been corrected

Line 142. Add “regression” after “Weighted quantile sum”

Thank you for the suggestion, this has been corrected in the statistical analysis

Line 143. Add “regression” after “Weighted quantile sum”

Thank you for the suggestion, this has been corrected in the statistical analysis

Line 187. Add “infection” before “admission” and change “admission” to admissions.

This has been corrected in the results section.

Line 195: change to “with $PM_{2.5} \leq 9 \mu g/m^3$, a $1-\mu g/m^3$ increase in $PM_{2.5}$ was associated....”

This has been added in the results section.

Lines 218-236: I think this would be better to just show in table 3, with some high level summary statements such as in line 223-224 “ $PM_{2.5}$ from oil combustion, coal burning and traffic were 224 associated with increased rate of hospital admission from intestinal infection”. Otherwise, there are a lot of long lists of numbers which are just repeated in table 3

Thank you for the suggestion. We’ve updated the results section correspondingly.

Lines 239-242: Long run on sentence. Suggest splitting to read “In this study among older adults across the contiguous US, we observed that higher ZIP-code level $PM_{2.5}$ concentrations were associated with increased admission rates from total and different subtypes of non-respiratory infection including intestinal infection, urinary tract infection and septicemia. The associations were observed even at lower levels of $PM_{2.5}$.”

Thank you for the suggestion. We’ve updated the discussion correspondingly.

Line 243: Awkward phrasing. Suggest rewording to “...and Cu contributed most to the association between...” Or “...and Cu had the highest percentage contribution to the association between...”

Thank you for the suggestion. We’ve rephrased the sentence.

Line 244: Change “Based on our sourced apportionment” to “Based on our source apportionment”

Thank you for the suggestion. We’ve corrected the sentence.

Line 245: Change “...and traffic were mostly associated...” to “...and traffic had the strongest associations with...”

This sentence in the discussion has been reworded.

Line 266: Unnecessary contraction, reword to “...infection have not...”

Thank you for the suggestion. This has been reworded.

Line 271: Reword to “...had the largest contribution to the effect...”

Thank you for the suggestion. This has been reworded.

Line 278: Reword to “...in rates and increase vulnerability to...”

Thank you for the suggestion. This has been reworded.

Line 279: Reword to “Exposure to high levels of Cu increases the...”

The sentence has been reworded.

Lines 290-291: Suggest changing to “It is possible” or “It is plausible”. I don’t think that you’ve established a likelihood for the sulfate interactions being the cause of the associations.

Thank you for the suggestion. The sentence has been updated.

Line 297: Something is wrong here. 2.7 million excess deaths in the U.S. would exceed total deaths in 2012. Perhaps you mean 2.7 million deaths globally?

Thank you for pointing this out. The number should be 0.36 million. We’ve corrected this in the manuscript.

Line 300: Reword to “Thurston et al observed that coal combustion PM2.5 is associated with increased mortality...”

Thank you for the suggestion. The sentence has been corrected.

Line 301: Reword to “...and that it showed a larger effect than total PM2.5 (44).”

Thank you for the suggestion. The sentence has been corrected.

Line 306: Reword to “Exhaust particles mainly come from tailpipe emissions while non-exhaust particles mainly come...”

Thank you for the suggestion. The sentence has been corrected.

Line 316: Reword to “...and thus may be less affected by...”

Thank you for the suggestion. The sentence has been reworded.

Line 318: Reword to “study is needed...” The adverb is not needed or justified.

Thank you for the suggestion. The sentence has been updated.

Lines 332-334. I would recommend also noting the potential for the use of zip codes with unequal areas and populations to lead to the potential for differential exposure measurement errors. I don't know how you can definitely state that the measurement error is likely to be non-differential.

Thank you so much for the suggestion. We agree that the measurement error may be differential and has added this to the limitation section.

Line 347: Reword to “Sulfates, nickel, and copper play the most important role...”

Thank you for the suggestion. The sentence has been updated.

Line 348: Reword to “...had larger effects on..”

Thank you for the suggestion. The sentence has been updated.

Line 350: Reword to “...PM2.5 on non-respiratory infections has been understudied and this study provides evidence that this pathway could be an important additional impact from PM2.5 and should be considered in evaluating the adequacy of current PM2.5 standards.”

Thank you for the suggestion. The conclusion has been updated.

Reviewer #2

Major revisions

- It is unclear how deaths were treated in the analyses. This should be clarified in the methods section. The sentence, “...were followed until the date of hospitalization from non-respiratory infections, death, or December 31st, 2016, whichever came earlier” suggests a time to event analysis. The analysis section specifies that a quasi-Poisson regression model was used but not how the model accounts for deaths. It is unclear if those who died after they were infected were included in the outcome counts of non-respiratory infections. It is unclear if the researchers were able to identify those who died from an infection without a prior hospitalization and whether these individuals were included in the non-respiratory infection counts.

Thank you so much for the suggestion. Instead of using time-to-event analysis, we aggregated the data of the beneficiaries by ZIP code and year and convert to count data. Each observation of the data included information on the number of hospitalizations from non-respiratory infection (count), total number of beneficiaries within a specific ZIP code during a specific year and the corresponding contextual variables of that ZIP code-year. Using quasi-Poisson regression, we modeled the number of hospitalization and using the number of beneficiaries as an offset. If a beneficiary died during a specific year, that patient will be excluded from the offset after that

year. (For example, if a beneficiary from a specific ZIP code died on 04/30/2007, the beneficiary would be included in the offset in the years during which he/she was enrolled in Medicare FFS, but being excluded from the offset in 2008 or later). We've further clarified this in the methods section of the manuscript. Those who died after they were infected were included in the outcome count as long as they were hospitalized before they died. Our data was not able to capture those who died from infection but not hospitalized. We understand that this could be a potential limitation in our study and have updated the limitation section. However, most of the death from infection came from the patient who had septicemia. Considering the severity of the disease, it's unlikely that patients would die from septicemia without being hospitalized. For intestinal infection and urinary tract infection, only small proportion of patients who were hospitalized ultimately died from infection. Therefore, we believe there were even fewer number of deaths among those who were not hospitalized.

- A more detailed description of the weighted sum quantile regression would help to clarify how the contribution of constituents were associated with the non-respiratory infection rates. It is unclear how the weighted sum quantile regression was combined with the quasi-Poisson model to estimate the relative contribution of each constituent on non-respiratory infection. There are no references or models provided for the weighted sum quantile regression and the appendix did not include a detailed discussion of this analysis.

Thank you for the suggestion. The generalized linear weighted quantile sum regression proceeds in steps. The original data is divided into two groups. Then each of the exposures is converted into a categorical variable representing the quantiles (in our case deciles) of that exposure. In the first group a regression (in our case quasi-Poisson) is fit relating the outcome to the quantiles of all of the exposures. Since all exposures are on the same scale (deciles) the coefficients of each exposure can be interpreted as the relative weight of a 1 decile increase in that exposure in predicting the outcome. The weights are scaled to sum to one. Then, using those weights, in the held-out data set a new variable is computed that is the weighted sum of each of the exposures (as deciles). This single variable is then used as the exposure in a regression (quasi-Poisson, in our case) predicting the outcome. Both regression control for all covariates. We have now provided these details in the appendix (supplementary methods).

- A more detailed explanation is provided for the source apportionment analysis, but it is spread between the manuscript and an appendix, which makes it difficult for researchers to understand and reproduce the analysis. Important details of the analysis are unclear. It would be helpful for the authors to include an overview of how the cluster analysis, non-negative matrix factorization, quasi-Poisson regression, and the random effect meta-analysis models were combined to obtain the estimates in Table 3. The authors do not provide details about the random effect meta-analysis models in either the manuscript or appendix.

Thank you so much for the suggestion. To illustrate this, we've included a flowchart (figure 1) in the manuscript. A more detailed description of the random effect meta-analysis has been provided in the updated appendix

Minor Comments

- It is unclear how the threshold for the most influential constituents was determined. In Figure S1 and S3, we see constituents with weighted concentrations very close to the threshold and they appear to be arbitrarily excluded or included as one of the most influential constituents.

The threshold for checking if one component's weight is significantly different from 0 is usually set at $(1/\text{number of species})$. The original paper discussing the criteria is Carrico C, Gennings C, Wheeler D, Factor-Litvak P. *Characterization of a weighted quantile sum regression for highly correlated data in a risk analysis setting. J Agricul Biol Environ Stat. 2014:1-21. ISSN: 1085-7117. DOI: 10.1007/s13253-014-0180-3. <http://dx.doi.org/10.1007/s13253-014-0180-3>. We've updated the manuscript and included that as a reference.*

- There are no references provided for the clustering algorithm. This approach is not commonly used so a reference or more detailed explanation is warranted.

Thank you for the suggestion. The manuscript is updated and the reference is included. We used Ward's hierarchical clustering because k-means clustering is computationally infeasible with ~30,000 ZIP codes. In addition, Ward's clustering provides an algorithm for choosing the number of clusters to stop at.

- Cardiovascular disease is a major cause of death and not specifically discussed in this manuscript. The authors could provide clarity by including references of studies exploring the relationship between PM2.5 and cardiovascular disease or a sentence on this relationship in the discussion.

Thank you for the suggestion. We've included reference for the association between PM2.5 and cardiovascular disease in the introduction (second paragraph).

Reviewer #3

Major comments:

The scale of the analysis needs clarification, especially early on in the manuscript, when describing the methods, and when putting the results in to context. For example, the time scale of the exposures is not specified in the abstract (e.g. are these results about daily increases in pollutants, annual increases in pollutants?). When describing the outcome variable in methods, there is good detail about how admissions are classified using ICD codes, but not about the how the infections are counted (e.g., annually? At the zip code level?) until the statistical analysis section. After reading the manuscript, I believe these results indicate that zip codes that experience higher annual averages of PM2.5 also experience higher hospital admissions for non-respiratory infections, but it is not explicitly stated. This comment is largely addressable.

Thank you so much for the suggestion. We've updated the manuscript correspondingly. The time scale has been specified in the abstract. Detailed description of how the infections were counted were included in the *outcome section*. We've also clarified our finding in the discussion and conclusion.

It is not clear how the autocorrelations between pollutant levels in zip codes across years, and

similarly the autocorrelations in hospital admissions by zip code across years, is handled. This is a serious concern. The analysis includes indicators for calendar year as covariates, but this does not appear to adequately address the lack of independence in observations from the same zip code across years. This calls the validity of the findings in to question.

Thank you for the comments. Autocorrelation biases estimates of standard errors, but not effect size estimates. Consequently, to account for the correlation, we incorporated robust standard error estimates in our analysis. The results in the manuscript are updated.

Similarly, how do findings handle the problem that the most populated places in the country (and therefore places most likely to have high hospital admissions) also appear to be among the most polluted (as suggested in the Figures)? More directly, it is not clear how much the results are from finding associations between increases in PM2.5 and hospital admissions in a given location, versus comparing hospital admissions between two locations that have different levels of PM2.5?

Thank you so much for the comment. While the most populated places tend to have higher admissions and air pollution, we use an annual population offset of the number of Medicare beneficiaries in that ZIP code in that year, so higher numbers of events in more populated locations are adjusted and will not confound the association. Differences in exposure within ZIP code between years as well as differences in exposure between locations both contribute to the variation in exposure that is correlated with the variation in counts of admissions. Besides the population offsets, the census level covariates, BRFSS, and Dartmouth health atlas variables also help control for confounding between locations, and the annual variations in those predictors help control for confounders that vary within location between years, along with the temperature variables. We agree that population density would be an important confounder in the association between PM2.5 and non-respiratory infection, since higher population density is associated with more pollution, and have included it as a confounder.

Grid cell level predicted exposure values for PM2.5 constituents combine predictions from random forests, gradient boosting, and neural networks (paragraph starting page 99). It's unclear how much sensitivity these predictions may have to the (multitude) of methods used. It's unclear how reliable/good estimates of annual PM2.5 exposure are to actual exposure levels, or how much they might vary given the variety of methods employed to estimate them. How sensitive are exposure estimates of PM2.5 to the (variety) of complex methods employed to estimate them? This is certainly an addressable comment.

Thank you for the comment. The exposure levels in this study were estimated from super learners which is a prediction algorithm that applies a set of candidate learners to the observed data and then combines them with another learner, trying several and choosing the optimal learner. Theoretically, the super learner will perform asymptotically as well or better than the

candidate learners (Sinisi, Sandra E., et al. "Super learning: an application to the prediction of HIV-1 drug resistance." *Statistical applications in genetics and molecular biology* 6.1 (2007).).

Total PM_{2.5} and PM_{2.5} constituents were estimated separately using different models. For the total PM_{2.5} level, daily level at each grid cell were first estimated using three machine learners (random forest, gradient boosting, and neural networks). For each grid cell, the final predicted daily PM_{2.5} level was obtained by combining the three estimates using generalized additive model. To assess how well these models performed, they were tested on held out data consisting of a subset of monitors not used in fitting the predictions. The predictions were compared to the actual exposure concentrations at these locations. This approach should evaluate the accuracy of the models. The final cross-validated R² was 0.86 for daily PM_{2.5} (comparing predicted to actual measured) and the cross-validated R² from the random forest, gradient boosting, and neural networks were 0.854, 0.818 and 0.855 respectively, suggesting combining the estimates from multiple machine learners improved the performance of the prediction model. Annual estimates of PM_{2.5} were estimated by averaging the daily PM_{2.5} level over the whole year. The cross-validated R² for annual PM_{2.5} was 0.89, suggesting that the modeled annual PM_{2.5} exposure was a reliable estimate to the actual exposure level. The cross-validated R² for the estimates of the 15 PM_{2.5} constituents ranged from 0.80-0.96. For each of the constituents, the super-learner outperformed the best candidate algorithm. None of this indicates whether an even better model might be obtained by using alternative learners, but the performance of the model is clearly good.

Minor comments:

Introduction: Would like more explanation of conflicting extant results of PM_{2.5} on non-respiratory infections (references 12-15). This could help strengthen the argument for the importance of this manuscript.

Thank you so much for this suggestion. We've updated the introduction and included the explanation.

The sentence beginning "To investigate the effect of PM_{2.5} at lower level, we conducted ..." is unclear. Lower level of what? Why is it important to do this?

Lower level indicates an annual PM_{2.5} concentration $\leq 9 \mu\text{g}/\text{m}^3$. We included this analysis because the US EPA is considering lowering the standard of annual PM_{2.5} (which is $12 \mu\text{g}/\text{m}^3$ currently) to $9-10 \mu\text{g}/\text{m}^3$. We conducted the analysis in order to find out if PM_{2.5} is still harmful at levels below the proposed standard. We've further clarified this in the statistical analysis section of the manuscript

Statistical analysis should include more support for the use of quasi Poisson models.

Thank you for the suggestion. We used quasi-Poisson regression because the outcome of the study is count data while the variance of the outcome is larger than the mean of outcome. We've clarified this in the statistical analysis section.

The scale of the reporting of hospital admissions increases for changes in PM2.5 concentrations (on the order 3%) is quite different from those for constituents (on the order of 10-20%). It would help to have more context on whether or not these are on similar 'scales,' or what a unit increase or sd increase actually means in terms of pollution.

In this analysis, effect of total PM2.5 was estimated for each 1 $\mu\text{g}/\text{m}^3$, while standard deviation of the total PM2.5 was around 3.7 $\mu\text{g}/\text{m}^3$. Suggesting that if the effect was estimated for 1 sd increase in PM2.5, the effect estimate would be around 11%, which was at a similar level to the constituents. The results have been updated and the effect for each standard deviation increase in PM2.5 are reported.

The sentence "Since the mental components ..." seems to belong in a limitations section, rather than in the first paragraph.

Thank you for the suggestion, we have moved this to the limitation section.

The authors should address how using Medicare data (largely older individuals'), may limit conclusions.

Thank you for the suggestion. We have updated the limitation section and discussed the potential limitation that brought by using data from older adults.

REVIEWERS' COMMENTS

Reviewer #1 (Remarks to the Author):

I appreciate the authors' response to my previous comments, especially regarding zip code measurement errors. I believe the authors have addressed my specific comments on the earlier draft.

Overall, the article still needs a good editorial review, specifically to fix plural nouns, e.g. in the title should be "Non-respiratory Infections", and in many cases hospital admission is used instead of hospital admissions. There are also a few other grammatical issues, for example in line 111, it should read "PM2.5 from oil combustion and coal burning were associated..." instead of "PM2.5 from oil combustion, coal burning were associated...". And there is an extra period at the end of line 114.

A couple of others, but note this is not a comprehensive list of all grammatical or spelling errors:

Line 117-118: Should read "Exhaust particles" instead of "Exhausted particles"

Line 208: Should be "arising" rather than "arrizing"

Reviewer #2 (Remarks to the Author):

This revision addressed my major concerns. I included a few minor comments below.

Minor comments:

Typo in Figure 4 in the main manuscript, 15 written twice in the second box.

The addition of the sections on weighted quantile sum regression and the random effects meta-analysis to the appendix help clarify what was done in the various stages of the analysis. However, as was done for the section of weighted quantile sum regression in the appendix, it would be helpful to include a reference in the appendix that provides more detailed information about random effects meta-analyses.

In Supplementary Figures S6-S14, it is unclear how the consensus shading was obtained. Please clarify this in the appendix or main manuscript.

Reviewer #3 (Remarks to the Author):

Overall comments:

The authors have been overall responsive to suggestions and should be congratulated for submitting a manuscript that is much easier to read and understand.

I have a few small comments, one comment that was only partly addressed in the revision. These are described below.

Overall, the results indicate that zip codes with higher levels of PM2.5 also have higher rates of hospital admissions for non-respiratory infections. The examination of non-respirator infections, as well as the effects of PM2.5 constituents and PM2.5 sources are additional to the literature.

Abstract

The abstract should include the number of zip codes and range of years.

Results

Lines 76-84. Some effects presented are per 1 unit increase in PM2.5, whereas others are per 3.7 unit increase. The authors explain the rationale in the rebuttal, but in the manuscript the switch/apparent inconsistency is confusing. I would recommend explaining why two different scales are used when presenting the results themselves.

Methods

From the rebuttal, it is clear that the authors did a great deal of data processing/prediction work to derive pm2.5 values and annual mean levels of constituents. However, in the manuscript, the description could still use clarification. It is unclear from the description if the authors implemented the variety of prediction methods themselves, or if they used data from that ensemble model that someone else implemented, or if they used standard software. The rebuttal mentions use of super learners, but this is not described in the

manuscript. Given that the conclusions depend heavily on deriving the exposure estimates, this is important to clarify.

A final small comment – it would help to justify the choice of three time periods for identification and classification of source-specific PM_{2.5}.

REVIEWERS' COMMENTS

Reviewer #1 (Remarks to the Author):

I appreciate the authors' response to my previous comments, especially regarding zip code measurement errors. I believe the authors have addressed my specific comments on the earlier draft.

Overall, the article still needs a good editorial review, specifically to fix plural nouns, e.g. in the title should be "Non-respiratory Infections", and in many cases hospital admission is used instead of hospital admissions. There are also a few other grammatical issues, for example in line 111, it should read "PM2.5 from oil combustion and coal burning were associated..." instead of "PM2.5 from oil combustion, coal burning were associated...". And there is an extra period at the end of line 114.

A couple of others, but note this is not a comprehensive list of all grammatical or spelling errors:

Line 117-118: Should read "Exhaust particles" instead of "Exhausted particles"

Line 208: Should be "arising" rather than "arrizing"

Thank you for the suggestions. I've updated the manuscript accordingly. The edits are highlighted in yellow

Reviewer #2 (Remarks to the Author):

This revision addressed my major concerns. I included a few minor comments below.

Minor comments:

Typo in Figure 4 in the main manuscript, 15 written twice in the second box.

Thank you for pointing out. The figure has been updated

The addition of the sections on weighted quantile sum regression and the random effects meta-analysis to the appendix help clarify what was done in the various stages of the analysis.

However, as was done for the section of weighted quantile sum regression in the appendix, it would be helpful to include a reference in the appendix that provides more detailed information about random effects meta-analyses.

Thank you for your suggestion. The reference for random effect meta-analysis has been added to the supplementary methods in the appendix

In Supplementary Figures S6-S14, it is unclear how the consensus shading was obtained.

Please clarify this in the appendix or main manuscript.

Figure S6-S14 shows the loadings of each constituent in different source factors obtained from non-negative matrix factorization. The constituents of PM_{2.5} mass usually originated from different sources. Specific constituents or groups of constituents that are highly associated with a source of air pollution could be serve as tracers, for example, Nickel and Vanadium are tracers of fuel oil combustion. We identified the source of each factor in figure S6-S14 through the loadings of the tracers. We've added the details to the appendix.

Reviewer #3 (Remarks to the Author):

Overall comments:

The authors have been overall responsive to suggestions and should be congratulated for submitting a manuscript that is much easier to read and understand.

I have a few small comments, one comment that was only partly addressed in the revision. These are described below.

Overall, the results indicate that zip codes with higher levels of PM_{2.5} also have higher rates of hospital admissions for non-respiratory infections. The examination of non-respirator infections, as well as the effects of PM_{2.5} constituents and PM_{2.5} sources are additional to the literature.

Abstract

The abstract should include the number of zip codes and range of years.

Thank you for the suggestion. These has been added to the abstract

Results

Lines 76-84. Some effects presented are per 1 unit increase in PM_{2.5}, whereas others are per 3.7 unit increase. The authors explain the rationale in the rebuttal, but in the manuscript the switch/apparent inconsistency is confusing. I would recommend explaining why two different scales are used when presenting the results themselves.

Thank you for pointing this out. We've updated the results section. All the results are presented for one standard deviation increase in PM_{2.5} (per 3.7 $\mu\text{g}/\text{m}^3$ increase) now.

Methods

From the rebuttal, it is clear that the authors did a great deal of data processing/prediction work to derive pm_{2.5} values and annual mean levels of constituents. However, in the manuscript, the description could still use clarification. It is unclear from the description if the authors implemented the variety of prediction methods themselves, or if they used data from that ensemble model that someone else implemented, or if they used standard software. The rebuttal mentions use of super learners, but this is not described in the manuscript. Given that the conclusions depend heavily on deriving the exposure estimates, this is important to clarify.

Thank you for the suggestion. The prediction models of our exposure were developed by other members of our group. The details of models have been published previously. We've clarified this in the manuscript and the reference are included. The super learner we mentioned in the rebuttal refers to ensemble machine learning models we mentioned in the manuscript, which incorporates the predictions from multiple machine learners.

A final small comment – it would help to justify the choice of three time periods for identification and classification of source-specific PM2.5.

Thank you for the suggestion. The regulation for the sources of air pollution vary overtime, at different time period, the sources are likely to be different. Therefore, we conducted the source apportionment separately for each time period. We chose three time periods because it allowed us to have enough power for source apportionment within each period while not sacrificing too much validity. We've updated the methods section in the manuscript to provide the reason for this.